# AniMer: Animal Pose and Shape Estimation using Transformer

## Abstract

Quantitative analysis of animal behavior and biomechanics requires accurate animal pose and shape estimation across species, and is important for animal welfare and biological research. However, the small network capacity of previous methods and limited multi-species dataset leave this problem underexplored. To this end, this paper presents **AniMer** to estimate ***ani***mal pose and shape using Trans-for***mer***, enhancing the reconstruction accuracy of diverse quadrupedal species. AniMer aims to unify the understanding of various quadrupedal forms within a single framework, overcoming the limitations of traditional methods that focus on narrow specific species. A key feature of AniMer is its integration of a high-capacity Transformer-based backbone, which significantly boosts performance. To effectively train AniMer, we aggregate most available open-sourced quadrupedal datasets, either with 3D or 2D labels, Further, we introduce Ctr-lAni3D, a novel large-scale synthetic dataset created through a diffusion-based conditional image generation pipeline, consisting of about 10k images with pixel-aligned SMAL labels. In total, we get 41.3k annotated images for training and validation. The combination of a robust backbone and an expansive dataset enables AniMer to outperform existing methods on the multi-species Animal3D dataset and several single-species dog benchmarks. Experiments on the unseen Animal Kingdom dataset further demonstrate the effectiveness of CtrlAni3D in enhancing the generalization ability of AniMer. Our study, through the development of AniMer and CtrlAni3D, underscores the significance of a large-capacity backbone and AI-driven synthetic data generation in advancing animal pose estimation research. Code and data will be released upon publication.

## 1 Introduction

Animal pose and shape estimation from images is essential for capturing animal behavior, biomechanics, and interactions with their environment, thereby yielding vital insights for animal welfare, agricultural practices and life sciences. The integration of geometric and appearance information from diverse species into a unified deep neural network represents a significant step to comprehensively interpret the intricate and dynamic animal world. A notable endeavor in this domain involves the estimation of pose and shape parameters from an articulated animal template known as the SMAL model (Zuffi et al., 2017), primarily targeting general quadrupeds. While extensive researches have been conducted regarding single species or families, such as horses (Zuffi et al., 2024; Li et al., 2021; Zuffi et al., 2019) and dogs (Sabathier et al., 2024; Rüegg et al., 2023; Rueegg et al., 2022; Li & Lee, 2021; Biggs et al., 2020; 2019), investigations into other quadrupedal species, including cats, cows or hippos, remain relatively underexplored.

The reconstruction of multiple species within a single network is hampered by the limited capacity of backbones and the scarcity of multi-species datasets annotated with SMAL labels. Recent advancements in human mesh recovery via the SMPL model (Loper et al., 2015) have demonstrated that using a high-capacity backbone in conjunction with large-scale datasets significantly enhances the accuracy of human pose and shape estimation in diverse settings (Goel et al., 2023; Pavlakos et al., 2024; Cai et al., 2024). However, this simple and effective paradigm remains untested within the context of animal studies. Xu et al. (2023a) introduced the first large-scale dataset named Animal3D featuring SMAL mesh annotations; nevertheless, their study predominantly utilized traditional CNN-based networks such as HMR (Kanazawa et al., 2018) and WLDO (Biggs et al., 2020).

In this paper, we present **AniMer**, a systematic approach pursuing accurate ***ani***mal pose and shape estimation using Transfor***mer***. AniMer surpasses existing methodologies on both the multi-species Animal3D dataset and several single-species dog benchmarks. This success rely on two key scaling factors: scaled backbone and scaled dataset. In previous animal researches, the well-known scaled Transformer backbone proposed by ViT (Xu et al., 2023b) has only been used for 2D animal pose estimation. We go beyond it by connecting ViT with a Transformer-based decoder to yield SMAL parameters, following the practice of HMR2.0 (Goel et al., 2023). To make such framework work for animals whose available data space and prior knowledge are limited compared to human, we modify the residual parameter decoding commonly used for human mesh recovery (Kanazawa et al., 2018; Goel et al., 2023) to direct parameter decoding, and adopt a two-stage training scheme to train on 3D dataset first instead of training on full dataset directly.

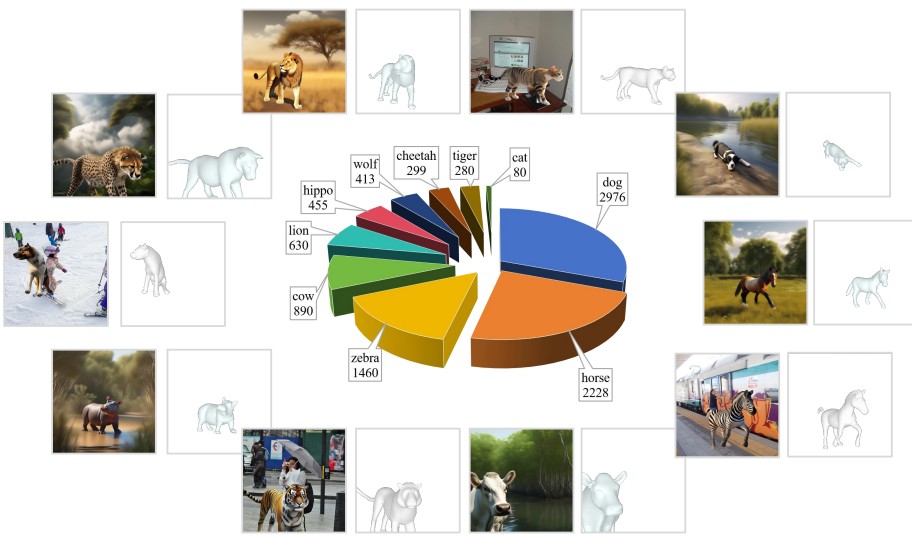

Figure 1: **CtrlAni3D dataset statistics.** For each image pair, the left side displays the generated animal image whose background comes from either COCO (Lin et al., 2014) or AI-synthesis, while the right side presents the rendering of the SMAL mesh label.

To obtain scaled dataset, we aggregate most available open-sourced quadrupedal datasets, resulting in a comprehensive set of 41.3k images annotated with either 3D mesh or 2D keypoints. Within the full expansive dataset, an important part is our newly proposed multi-species 3D animal dataset termed CtrlAni3D. It is a synthetic dataset generated by a novel rendering pipeline that extracts AI knowledge regarding animal geometry and appearance from a diffusion-based conditional image generation methodology. Specifically, we prompt ControlNet (Zhang et al., 2023) with textual descriptions of animal behaviors and rendered SMAL mask and depth images, resulting in highly realistic visual outputs, as illustrated in Fig. 1. To ensure dataset quality, we use SAM2.0 (Ravi et al., 2024) together with manual verification to filter implausible generated images. Finally, CtrlAni3D comprises 9711 images annotated with pixel-aligned SMAL mesh, and its scalable nature allows for further expansion. In contrast to traditional CG pipelines for synthetic data generation (Cao et al., 2019; Xu et al., 2023a), CtrlAni3D exhibits superior rendering quality and reduced labor intensity.

To rigorously evaluate the efficacy of the AniMer model and the CtrlAni3D dataset, we conduct extensive empirical studies. Our findings indicate that AniMer, when trained on the same full multi-species datasets, significantly outperforms CNN-based methods such as HMR and WLDO (Biggs et al., 2020), which serve as baselines in the Animal3D research. Notably, AniMer demonstrates improved performance over previous methods on dog pose and shape estimation, emphasizing its representation ability on single species tasks. Through comprehensive ablation studies, we verify that CtrlAni3D enhances the generalization abilities of AniMer on the unseen Animal Kingdom dataset (Ng et al., 2022).

In conclusion, AniMer represents a substantial advancement toward a generalized 3D understanding of animals, characterized by its simple yet effective design principles. Our contributions are

summarized in three key aspects: firstly, we introduce AniMer, the first Transformer-based method for animal pose and shape estimation, illustrating the efficacy of high-capacity network across multiple animal species; secondly, we propose CtrlAni3D, a novel large-scale synthetic dataset generated via a conditional image generation model, with a scalable pipeline applicable to a broader range of species; lastly, AniMer demonstrates remarkable performance improvements for general quadrupedal animals, and experiments on dogs further affirm its superiority in pose and shape estimation for specific species. Code and data will be release upon publication.

## 2 RELATED WORKS

**Animal Pose and Shape Estimation.** In this paper, we concentrate on template-based methods instead of template-free methods (Yang et al., 2022a; Yao et al., 2022; Jakab et al., 2024; Li et al., 2024b). Reconstructing the shape and pose of animals has been studied on various kinds of animal families such as birds (Wang et al., 2021), mouse (An et al., 2023), non-human primates (Neverova et al., 2020) and quadrupeds (Zuffi et al., 2018). Although we focus on quadrupeds in this paper, the proposed strategy could apply to other animal families. For quadrupeds, Zuffi et al. (2017) proposes a well-known parametric model SMAL, which is built upon 41 scans of animal toys. Due to the limited geometry accuracy of SMAL for representing specific species, most previous methods stride over predicting parametric SMAL parameters only and further enhance the geometry of horses and dogs. For example, Li et al. (2021) designs the hSMAL model specific to horse and applies it to the problem of lameness detection from video. VAREN (Zuffi et al., 2024) further improves hSMAL with super high quality horse scans. Similarly, using SMAL for dog mesh recovery is addressed by adding bone lengths control (Biggs et al., 2020), by adding per-vertex deformation (Li & Lee, 2021), by introducing breed loss (Rueegg et al., 2022), by ground contact constraints (Rüegg et al., 2023), or by temporal avatar optimization (Sabathier et al., 2024). Though super high quality reconstruction has been achieved on horses and dogs, end-to-end SMAL estimation has not been fully investigated on other challenging species. Animal3D (Xu et al., 2023a) provides the first large scale 3D benchmark for general quadruped SMAL estimation, yet neglects in-depth research on the network itself. Therefore, our proposed AniMer delves into the network design choice of SMAL recovery, and could be integrated with different geometry enhancing methods summarized above to further reduce geometric errors.

**Transformer Based Human Mesh Recovery.** The Transformer architecture (Vaswani, 2017) has revolutionized the field of Natural Language Processing (NLP) by enabling unprecedented accuracy and efficiency in a wide range of tasks. Inspired by its success in NLP, one of the core part of Transformer, i.e. self-attention, has been widely used for human mesh recovery (Kocabas et al., 2021; Wan et al., 2021; Shen et al., 2023; Shin et al., 2024). Dosovitskiy (2020) proposes Vision Transformer(ViT), which divides an image into patches as the input to Transformer. ViT has achieved state-of-the-art performance on several computer vision tasks, including human mesh recovery using SMPL (Kocabas et al., 2021; Wan et al., 2021; Shen et al., 2023; Goel et al., 2023; Shin et al., 2024). Among all these works, HMR2.0 (Goel et al., 2023) is a milestone which demonstrates the effectiveness of simply using ViT backbone and large scale datasets to achieve accurate mesh recovery and in-the-wild generalization ability. Inspired by this, HaMeR (Pavlakos et al., 2024) employs ViT backbone to achieve highly accurate hand mesh recovery, which is further extended to interacting hands (Lin et al., 2024). Similarly, SMPLer-X (Cai et al., 2024) scales up expressive human pose and shape estimation using ViT backbone and the combination of 32 datasets. Although impressive results have been achieved in human mesh recovery, the effect of ViT backbone for animal pose and shape estimation remains unexplored.

**Synthetic Animal Training.** Compared to human pose estimation which benefits from large-scale datasets, acquiring annotated images of animals is significantly more difficult. Therefore, synthetic dataset would alleviate this problem by rendering the input and output simultaneously. However, all previous methods only attempt to render RGB images or depth images using traditional computer graphics pipelines, ignoring the image hallucination ability of generative AI models such as stable diffusion (Rombach et al., 2022) or ControlNet (Zhang et al., 2023). Specifically, Mu et al. (2020) utilizes animal CAD models to generate animal images with 2D keypoints and part segmentation for training. RGBD-dog (Kearney et al., 2020) synthesizes depth images with labels to train depth-based 3D skeleton prediction network. BITE (Rüegg et al., 2023) builds semi-synthetic 3D test dataset based on scans of real dogs. DigiDogs (Shooter et al., 2024) generates outdoor dog videos

together with GT labels using Grand Theft Auto (GTA) game engine. Different from above methods which only focus on single species or family, (Xu et al., 2023a) renders multi-species images using textured SMAL meshes with vertex deformations obtained by SMALR (Zuffi et al., 2018). Although above traditional CG rendering achieves success in assisting network training, the plausibility of rendered images is hindered by the coarse texture quality and sophisticated lighting and shadow control.

# 3 RECONSTRUCTING ANIMALS USING ANIMER

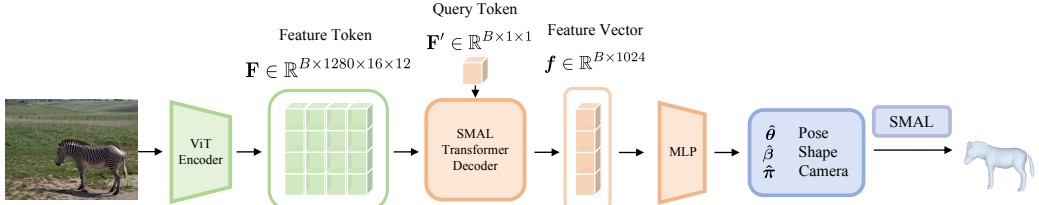

Figure 2: **AniMer Network Architecture.** AniMer consists of (1) a ViT encoder that extracts image features; (2) a transformer decoder that processes the image features from the encoder; (3) MLPs which regress shape $\hat{\beta}$, pose $\hat{\theta}$ and camera parameter $\hat{\pi}$.

## 3.1 PRELIMINARIES

**The SMAL Model.** The SMAL model, denoted as $\mathcal{M}(\beta, \theta, \gamma)$, is a 3D parametric shape model designed for quadrupeds. The shape parameter $\beta \in \mathbb{R}^{41}$ is derived from 41 3D scans of various animal figurines, including cats, dogs, horses, cows, and hippos (Zuffi et al., 2017). The pose parameter $\theta \in \mathbb{R}^{35 \times 3}$ represents the rotation of each joint relative to its parent joint, expressed in terms of axis-angle. Controlled by $\beta$ and $\theta$, the SMAL model outputs a mesh with vertices $V \in \mathbb{R}^{3889 \times 3}$ and faces $F \in \mathbb{N}^{7774 \times 3}$ through linear blend skinning (LBS) process. The animal body joints are regressed from vertices by $J \in \mathbb{R}^{35 \times 3} = W \cdot V$, where $W \in \mathbb{R}^{35 \times 3889}$ represents a linear mapping matrix.

**Camera Projection.** Following Goel et al. (2023), $\pi(\cdot)$ represents the projection process of a weak-perspective camera model, which is determined by a translation vector $T \in \mathbb{R}^3$, a fixed focal length $f = 1000$ and thereby a fixed intrinsic matrix $K$. The global orientation $R$ is the rotation of root joint, therefore we ignore it here. Consequently, a 3D point $X$ is projected as 2D point $x$ by $x = \pi(X) = \Pi(K(X + T))$, where $\Pi$ is typical perspective projection procedure.

## 3.2 THE ARCHITECTURE OF ANIMER

The full architecture of AniMer is shown in Fig. 2. Specifically, given an image $I$, we first utilize a ViT encoder to extract image feature tokens $\mathbf{F}$. We then feed the feature tokens $\mathbf{F}$ into a SMAL transformer decoder to obtain a feature vector $\boldsymbol{f}$. Finally, independent multi-layer perceptrons (MLPs) are used to predict the shape parameter $\hat{\beta}$, pose parameter $\hat{\theta}$, and the camera parameter $\hat{\pi}$, where $\hat{\cdot}$ means estimated parameters. Note that the weights of ViT encoder are pre-trained on ImageNet (Deng et al., 2009) using Masked Autoencoders (MAE) (He et al., 2022), which significantly enhances mesh recovery performance.

To make such Transformer-based structure works well for animals, AniMer features two key differences compared with previous Transformer-based human or hand mesh recovery methods. First, different from HMR2.0 (Goel et al., 2023) and HaMeR (Pavlakos et al., 2024) which regress the residual SMPL/MANO parameters with respect to the non-zero mean parameters computed from large scale motion databases, we choose to directly decode the final SMAL parameters due to limited SMAL pose prior. Second, both HMR2.0 and HaMeR train on the whole datasets in single stage. Instead, we train AniMer in two stages. In the first stage, we train AniMer using only 3D data to ensure the network could predict feasible shapes and poses. In the second stage, all 3D and 2D data are introduced for training. The insight is that the size of 3D datasets for animal is much smaller

than that of human at present, resulting in an imbalanced 3D and 2D data scale. Unless otherwise specified, we train two stages for 500 and 700 epochs respectively.

## 3.3 LOSS FUNCTIONS

To align animal images with reconstruction results, we train our model using a comprehensive loss function that incorporates various 2D and 3D annotations. We define the main loss function $\mathcal{L}_{\text{total}}$ as a weighted sum of several loss components, each focusing on different aspects of the model's performance. The main loss function is given by

$$\mathcal{L}_{\text{total}} = \lambda_{3D}\mathcal{L}_{3D} + \lambda_{2D}\mathcal{L}_{2D} + \lambda_{\text{prior}}\mathcal{L}_{\text{prior}} + \lambda_{\text{adv}}\mathcal{L}_{\text{adv}}. \tag{1}$$

Here, $\lambda_{3D} = 0.05, \lambda_{2D} = 0.01, \lambda_{\text{prior}} = 0.001, \lambda_{\text{adv}} = 0.0005$ are the loss weights. For 3D training data, all losses are used. For samples without 3D annotations, the 3D loss $\mathcal{L}_{3D}$ is disabled.

**3D Loss.** For images annotated with SMAL model parameters $\beta$ and $\theta$, we supervise these parameters directly to enable fast convergence. Additionally, we also supervise the estimated 3D keypoints $\hat{K}_{3D}$ with ground truth $K_{3D}$ to assist in better 3D joint localization. The 3D loss function is then defined as

$$\mathcal{L}_{3D} = \lambda_{\beta}||\hat{\beta} - \beta||_2^2 + \lambda_{\theta}||\hat{\theta} - \theta||_2^2 + ||\hat{K}_{3D} - K_{3D}||_1, \tag{2}$$

where $\lambda_{\beta} = 0.01$ and $\lambda_{\theta} = 0.2$ are loss weights. The term $|| \cdot ||_2^2$ denotes squared L2 norm, while $|| \cdot ||_1$ represents L1 norm.

**2D Loss.** Because most training data only contain 2D-level annotations (2D keypoints or masks, yet masks are for evaluation only). For these data, we supervise 2D keypoints during training using $\mathcal{L}_{2D} = ||\pi(\hat{K}_{3D}) - K_{2D}||_1$.

**Prior Loss.** Furthermore, to ensure that the predicted shape and pose parameters are natural, we enforce them to be close to a prior distribution by calculating the Mahalanobis distance. The prior loss is defined as

$$\mathcal{L}_{\text{prior}} = \lambda_{\beta}(\hat{\beta} - \mu_{\beta})^T\Sigma_{\beta}^{-1}(\hat{\beta} - \mu_{\beta}) + (\hat{\theta} - \mu_{\theta})^T\Sigma_{\theta}^{-1}(\hat{\theta} - \mu_{\theta}), \tag{3}$$

where $\lambda_{\beta} = 0.5, \mu_{\beta}, \Sigma_{\beta}, \mu_{\theta}$, and $\Sigma_{\theta}$ are the mean and covariance of the prior distributions given by SMAL (Zuffi et al., 2017).

**Adversarial Loss.** Finally, we adopt a discriminator to further ensure that the model outputs natural poses and shapes, therefore we employ an adversarial loss similar to HMR (Kanazawa et al., 2018). This loss is designed to make the predicted parameters indistinguishable from real distribution. It is defined as $\mathcal{L}_{\text{adv}} = \sum_k(D_k(\theta, \beta) - 1)^2$, where $D_k$ represents a discriminator.

## 4 CTRLANI3D DATASET

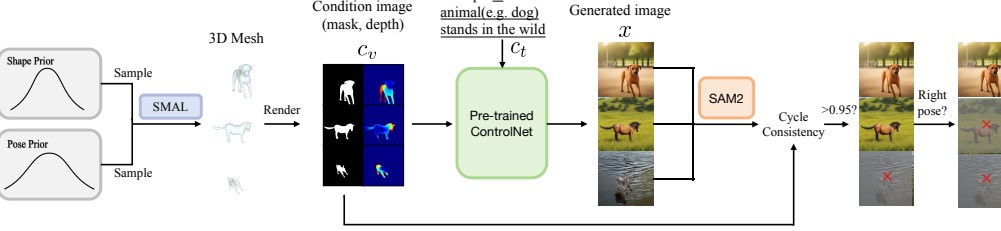

Figure 3: **Pipeline for CtrlAni3D generation.** We begin by randomly sampling shapes and poses from prior spaces to generate SMAL mesh. Then we render SMAL mesh into mask and depth map, which serve as structural conditional images. We further combine information about animal species, pose, and behavior into a prompt. The prompt, along with the conditional images, are fed into a pretrained ControlNet to generate images. Finally, we utilize SAM2 to segment the foreground of generated images and calculate Intersection-over-Union (IOU) between rendered mask and segmented mask as a cycle consistency constraint. Only when IOU exceeds 0.95, the image could be sent for manual verification.

**Motivation.** Acquiring a large number of animal images with full 3D annotations is difficult, and synthetic datasets can supplement the shortcomings of real data. Previous methods use traditional graphics pipeline to render either deformed SMAL models obtained by fitting images (Xu et al., 2023a) or manually created CG assets (Shooter et al., 2024) to obtain image-label pairs. However, the plausibility of rendered images is hindered by the coarse texture quality and sophisticated lighting and shadow control. These limitations motivate our usage of ControlNet (Zhang et al., 2023) to create high quality AI hallucinated images conditioned on SMAL structures. This principle effectively bridges the domain gap between parametric labels and images, and help us to create a novel large scale dataset named CtrlAni3D with minimal human labor involved.

**SMAL Structure Condition.** The dataset generation pipeline is illustrated in Fig. 3. Given a posed SMAL mesh and viewpoint, we render it into mask map and depth map as the condition images $c_v$ to guide the structure of images $x$ generated by a pre-trained ControlNet. To guarantee the diversity of the poses and shapes used for synthesis, we randomly sample $\beta$ from the Gaussian distributed shape space provided by original SMAL (Zuffi et al., 2017), and sample a more diverse range of $\theta$ from a combined pose space presented by dog models (Rüegg et al., 2023; Li et al., 2024a; Biggs et al., 2020). This is reasonable because the quadrupeds expressed by SMAL share similar anatomical structures. About the viewpoint for rendering, each dimension of the global rotation vector is uniformly sampled from $(-\pi, \pi)$ while the position is uniformly sampled between $[-0.5, -0.5, 4]$ and $[0.5, 0.5, 8]$.

**Text Condition.** To further control the style of generated images $x$, we seek to use text prompt. We first manually classify the sampled 3D meshes into 10 species: cat, tiger, lion, cheetah, dog, wolf, horse, zebra, cow and hippo, and use species name as one keyword. The second keyword is the pose description, e.g. "stands" in Fig. 3, which is assigned by human annotator according to the 3D mesh. Based on these keywords, ChatGPT (Achiam et al., 2023) is employed to complete a prompt sentence $c_t$ depicting possible animal behaviors. Finally, both $c_v$ and $c_t$ act as the prompts of ControlNet for realistic and rich image generation.

**Semi-Automated Filtering.** Note that, not all generated images are perfectly aligned with conditions. To address this, we design a semi-automated filtering strategy to lower the burden of annotators. First, SAM2 (Ravi et al., 2024) is utilized to extract the foreground mask of the generated images, enabling cycle-consistency checking by comparing to conditioned mask. Further, we manually filter out images that do not match the mesh poses to ensure good data quality. Each synthetic image has a resolution of $512 \times 512$ pixels and includes well-aligned annotations for $\beta$, $\theta$, $\gamma$ and 3D keypoints. By comparing rendered depth image the projected keypoint depths, we also obtain visible 2D keypoints as annotation.

# 5 EXPERIMENTS

## 5.1 SETUP

**Datasets.** In this paper, we curate and aggregate multiple datasets containing 2D and 3D annotations for animals. Specifically, the full dataset includes Animal Pose (Cao et al., 2019), APT-36K (Yang et al., 2022b), AwA-Pose (Banik et al., 2021), Stanford Extra (Biggs et al., 2020), Zebra synthetic (Zuffi et al., 2019), Animal3D (Xu et al., 2023a), and our own CtrlAni3D. For evaluation, we mainly use the test part of Animal3D and CtrlAni3D, and the unseen Animal Kingdom dataset (Ng et al., 2022). Note that, only 8 quadruped species from Animal Kingdom are selected for testing. For additional experiments on single-species dog mesh recovery task, we evaluate the network on the test splits of Stanford Extra, Animal Pose, Animal3D, and Cop3D (Sinha et al., 2023). For more details about the datasets, please refer to the Appendix.

**Training Details.** Our model is implemented by Pytorch Lightning. We use AdamW (Loshchilov & Hutter, 2019) optimizer with a linear learning rate decay schedule. The initial learning rate is $1.25 \times 10^{-6}$. The entire training takes 80 hours on a NVIDIA RTX 4090 GPU.

**Metrics.** Several 3D and 2D metrics are employed to fully assess the model performance, listed below. **PA-MPJPE** is Procrustes-Aligned Mean Per Joint Position Error (PA-MPJPE) for regressed 3D keypoints. **PA-MPVPE** is Procrustes-Aligned Mean Per Vertex Position Error (PA-MPVPE) for SMAL vertices. **IOU** is Intersection Over Union for mask comparison. It compares how well

the rendered masks align with ground truth masks. **IOUw5** is IOU over the 5% worst performed samples. **PCK** is Percentage of Correct Keypoints given a threshold. In this paper, PCK is only used for evaluating 2D keypoints. PCK@HTH uses half the head-to-tail distance as the threshold. By setting threshold to 0.1 and 0.15, we get commonly used PCK@0.1 and PCK@0.15 metrics. **AUC** is Area Under the Curve value when the PCK threshold gradually increase from 0 to 1.

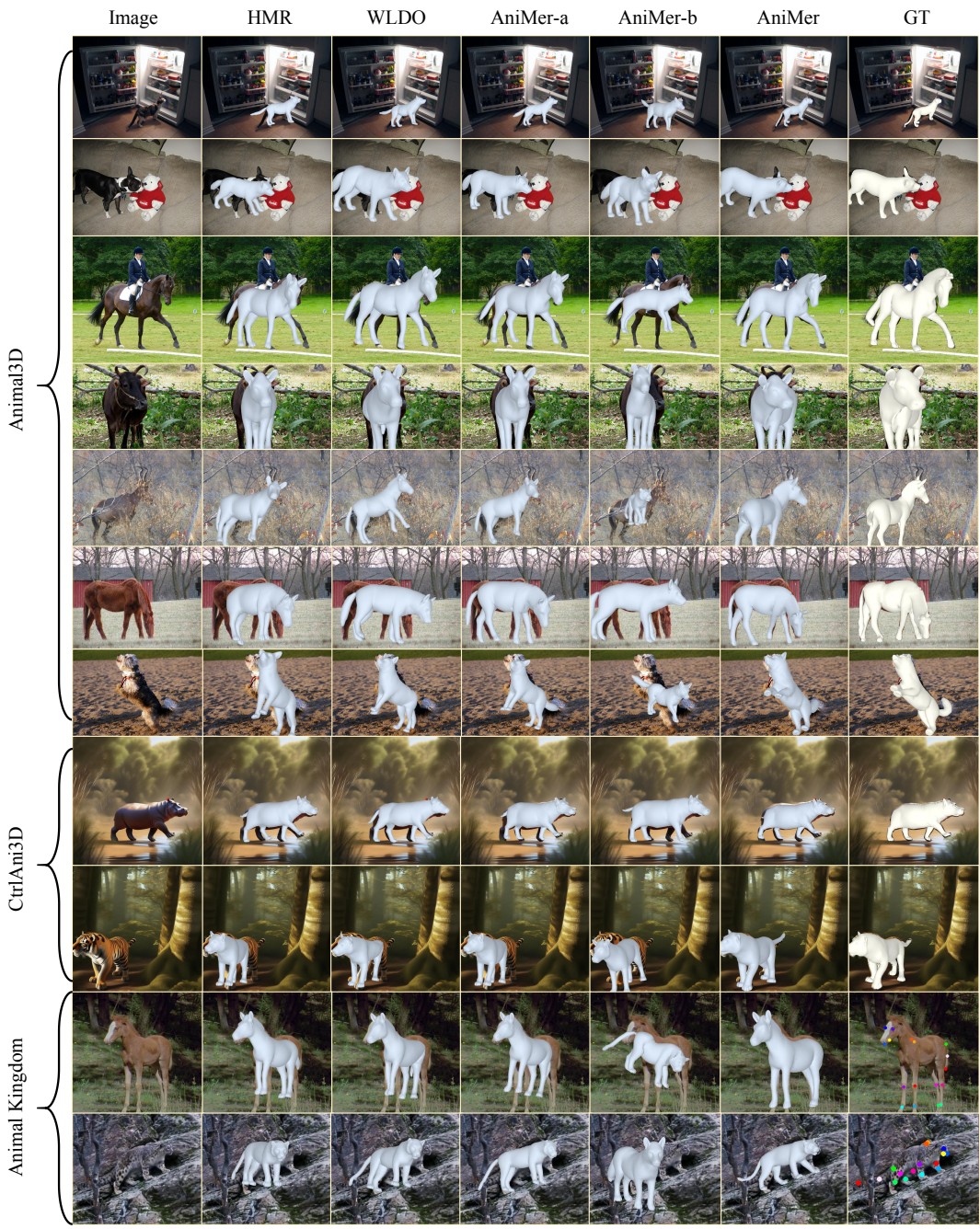

Figure 4: **Qualitative comparisons on Animal3D, CtrlAni3D and Animal Kingdom datasets.** We compare our results with HMR (Kanazawa et al., 2018), WLDO (Biggs et al., 2020), AniMer-a (ResNet152 backbone), and AniMer-b (no MAE pretraining). On a wide range of animal species, AniMer achieves better image-aligned shape and pose estimation.

## 5.2 MULTI-SPECIES EXPERIMENTS

Following (Xu et al., 2023a), we utilize HMR (Kanazawa et al., 2018) and WLDO (Biggs et al., 2020) as baselines. Furthermore, to highlight the significance of our ViT backbone and backbone pretraining, we compare the final AniMer with two variants: AniMer-a which replaces ViT backbone with ResNet152, and AniMer-b which discards MAE pretraining. To be fair, all above methods are trained on our full dataset with the same losses and the same two-stage training strategies.

Quantitative results are shown in Tab. 1. We find that WLDO is the previously best method while AniMer performs significantly better. On Animal3D dataset, AniMer promotes over WLDO by 14%, 31%, 28% and 16% for AUC, PCK, PA-MPJPE and PA-MPVPE respectively. Besides, on CtrlAni3D dataset, AniMer outperforms WLDO by 6%, 9%, 45% and 49% for AUC, PCK, PA-MPJPE and PA-MPVPE metrics. AniMer performs well not only on 3D datasets, but also on in-the-wild 2D dataset that is never seen during training. Specifically on Animal Kingdom dataset, AniMer demonstrates strong robustness and improves over WLDO by 16.2% for AUC. Qualitative results in Fig. 4 shows that AniMer aligns with images much better for thin structures such as legs and tails. More qualitative results are shown in Appendix.

Table 1: **Quantitative comparisons on Animal3D, CtrlAni3D and AnimalKingdom datasets. Bold** numbers indicate the best performance within the same evaluation set.

| Dataset | Metric | Method | | | | |
|---|---|---|---|---|---|---|
| | | HMR | WLDO | AniMer-a | AniMer-b | AniMer |
| Animal3D | AUC↑ | 76.3 | 78.2 | 75.2 | 60.6 | **89.1** |
| | PCK@HTH↑ | 60.8 | 68.7 | 57.2 | 38.9 | **89.9** |
| | PA-MPJPE↓ | 123.5 | 112.3 | 115.5 | 147.9 | **81.2** |
| | PA-MPVPE↓ | 133.9 | 125.2 | 128.7 | 157.6 | **85.3** |
| CtrlAni3D | AUC↑ | 80.8 | 88.7 | 80.3 | 78.5 | **93.9** |
| | PCK@HTH↑ | 67.0 | 86.7 | 66.0 | 65.9 | **95.3** |
| | PA-MPJPE↓ | 123.5 | 71.5 | 117.0 | 102.3 | **39.4** |
| | PA-MPVPE↓ | 133.9 | 83.4 | 129.4 | 112.6 | **42.7** |
| Animal Kingdom | AUC↑ | 70.2 | 70.1 | 68.9 | 45.4 | **86.3** |
| | PCK@HTH↑ | 64.0 | 64.3 | 62.5 | 31.8 | **85.2** |
| | PCK@0.1↑ | 12.8 | 14.6 | 10.2 | 4.0 | **37.4** |
| | PCK@0.15↑ | 25.6 | 27.6 | 21.3 | 9.2 | **57.2** |

## 5.3 SINGLE-SPECIES (DOG) EXPERIMENTS

In addition, to demonstrate the effectiveness of AniMer for reconstructing specific species, we compare with state-of-the-art dog mesh recovery methods WLDO (Biggs et al., 2020), Coarse-to-fine (Li & Lee, 2021), BARC (Rueegg et al., 2022), BITE (Rüegg et al., 2023) and Animal Avatar (Sabathier et al., 2024). For a fair comparison, we train AniMer using the same dataset protocol as used in previous methods (Rüegg et al., 2023). Note that Coarse-to-fine uses GCN to deform SMAL for improved joint and vertex positions while other methods all use original SMAL geometry. Besides, BITE incorporates a test-time-optimization (ttopt) and Animal Avatar incorporates appearance-based spatiotemporal optimization. For fair comparison with BITE, we also report our results with ttopt strategy in Tab. 2. Moreover, without ttopt, our method still achieves a better IOU of 84.7 on Stanford Extra while BITE achieves 84.2. The IOUw5 of ours is 47.6 compared to the 45.3 of BITE, indicating more robust results of our method over challenging cases. Qualitative results of both with and without ttopt for BITE and our method are shown in Fig. 5.

Tab. 2 also shows that our method outperforms all existing methods on 2D datasets Stanford Extra and Animal Pose, yet performs competitive to BITE on Animal3D dataset. This indicates that our method generalizes better to datasets without 3D supervision. On the challenging pet video dataset Cop3D, our method achieves the best performance among per-frame approaches, but Animal Avatar performs better on IOUw5 metric. The reason is that IOUw5 accounts for the worst frames while Cop3D contains frames with extreme viewpoints and highly truncated targets. The joint optimization across the entire time series of Animal Avatar makes it more robust to these extreme cases.

Table 2: **Quantitative comparisons for dogs**. "ttopt" represents test-time-optimization. "*" means the numbers on Cop3D dataset are borrowed from Sabathier et al. (2024) except Ours.

| Method | Stanford Extra | | Animal Pose | | Animal3D | | | Cop3D* | |
|---|---|---|---|---|---|---|---|---|---|
| | PCK↑ | IOU↑ | PCK↑ | IOU↑ | PCK↑ | IOU↑ | PA-MPJPE↓ | IOU↑ | IOUw5↑ |
| WLDO | 74.2 | 78.8 | 67.6 | 67.5 | 52.1 | 66.7 | 157.8 | 77.4 | 54.6 |
| Coarse-to-fine | 81.6 | 83.4 | 67.8 | 75.7 | 61.9 | 76.8 | 147.2 | 82.5 | 64.9 |
| BARC | 75.1 | 82.8 | 65.9 | 67.5 | 61.9 | 71.4 | 112.4 | 75.0 | 47.0 |
| BITE(ttopt) | 85.8 | 85.2 | 71.1 | 82.6 | 66.6 | **81.3** | 107.9 | 81.0 | 59.0 |
| Animal-Avatar | - | - | - | - | - | - | - | 84.0 | **79.0** |
| Ours(ttopt) | **86.4** | **85.6** | **71.8** | **82.8** | **67.9** | 81.1 | **105.8** | **86.6** | 69.3 |

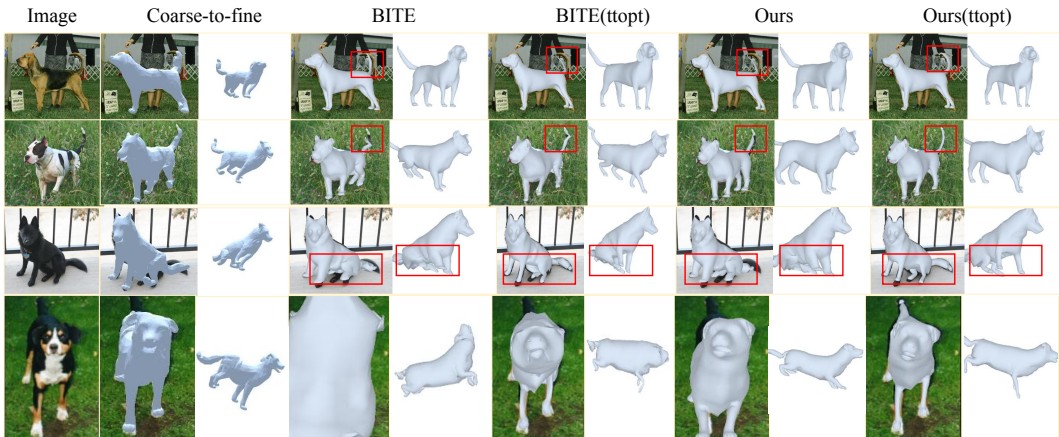

Figure 5: **Qualitative comparisons on Stanford Extra dataset.** "ttopt" represents test-time-optimization. Each row displays the input image (left) and results from different methods (right). For each result, a overlapped image and a side view rendering are shown. The red boxes indicate that our method is more accurate than BITE pose estimation and has better reconstruction details.

## 5.4 EFFECT OF CTRLANI3D

To further demonstrate the effectiveness of our proposed CtrlAni3D dataset in enhancing network capabilities, we conduct a series of ablation experiments. We compare models trained on the original Animal3D dataset (A3D), A3D combined with our CtrlAni3D dataset (C3D), and full 3D and 2D datasets. The hyper parameters of training are kept the same. By testing the performance on datasets with 3D ground truth, Tab. 3 shows that CtrlAni3D itself improves the model accuracy not only on CtrlAni3D testset, but also Animal3D testset. Furthermore, the incorporation of 2D datasets further improve the performance by a large margin, proving that our large-scale training set collection is one of the key to robust multi-species animal pose and shape estimation.

Table 3: **Effect of CtrlAni3D on 3D evaluation datasets.** We report PA-MPJPE and PA-MPVPE (in mm) as 3D metrics, and AUC with PCK@HTH (in percentage) as 2D metrics. **Bold** numbers indicate the best performance within the same evaluation set.

| Evaluation Dataset | Training Data | | | 3D Metric | | 2D Metric | |
|---|---|---|---|---|---|---|---|
| | A3D | C3D | 2D | PA-MPJPE↓ | PA-MPVPE↓ | AUC↑ | PCK@HTH↑ |
| Animal3D | ✓ | | | 87.3 | 93.2 | 86.3 | 83.6 |
| | ✓ | ✓ | | 82.5 | 88.0 | 88.0 | 88.2 |
| | ✓ | ✓ | ✓ | **81.2** | **85.3** | **89.1** | **89.9** |
| CtrlAni3D | ✓ | | | 90.1 | 95.8 | 88.8 | 88.8 |
| | ✓ | ✓ | | 54.6 | 59.2 | 92.8 | 95.1 |
| | ✓ | ✓ | ✓ | **39.4** | **42.7** | **93.9** | **95.3** |

Table 4: **Effect of CtrlAni3D on 2D evaluation datasets.** "*" means that only dog species are used for Animal Pose and Stanford Extra. We report AUC, PCK@HTH, PCK@0.1 and PCK@0.15 as 2D metrics. **Bold** numbers indicate the best performance within the same evaluation set.

| Evaluation Dataset | Training Data | | | 2D Metric | | | |
|---|---|---|---|---|---|---|---|
| | A3D | C3D | 2D | AUC↑ | PCK@HTH↑ | PCK@0.1↑ | PCK@0.15↑ |
| Animal Kingdom | ✓ | | | 80.1 | 78.4 | 25.9 | 46.2 |
| | ✓ | ✓ | | 81.4 | 81.1 | 30.4 | 48.9 |
| | ✓ | ✓ | ✓ | **83.6** | **85.2** | **37.4** | **57.2** |
| Animal Pose* | ✓ | | | 82.1 | 85.7 | 25.4 | 48.2 |
| | ✓ | ✓ | | 83.9 | 88.8 | 36.2 | 57.7 |
| | ✓ | ✓ | ✓ | **87.2** | **92.8** | **48.0** | **69.4** |
| Stanford Extra* | ✓ | | | 84.7 | 90.1 | 39.6 | 64.1 |
| | ✓ | ✓ | | 86.6 | 92.4 | 49.8 | 73.0 |
| | ✓ | ✓ | ✓ | **89.3** | **94.8** | **66.3** | **84.2** |

Similarly, ablation studies on the unseen Animal Kingdom dataset further validates the value of CtrlAni3D and full dataset collection for multi-species mesh recovery. Specifically, Tab. 4 indicates that our model, when trained with CtrlAni3D, exhibits superior performance even on previously unseen in-the-wild data. Qualitative comparisons in Fig. 6 further demonstrates that training with CtrlAni3D helps AniMer to yield more accurate animal terminal body parts such as tails, limbs and faces. Furthermore, on dog-specific datasets such as Animal Pose (only dog part) and Stanford Extra, CtrlAni3D also improves the model performance though trained for multi-species tasks, see bottom rows of Tab. 4.

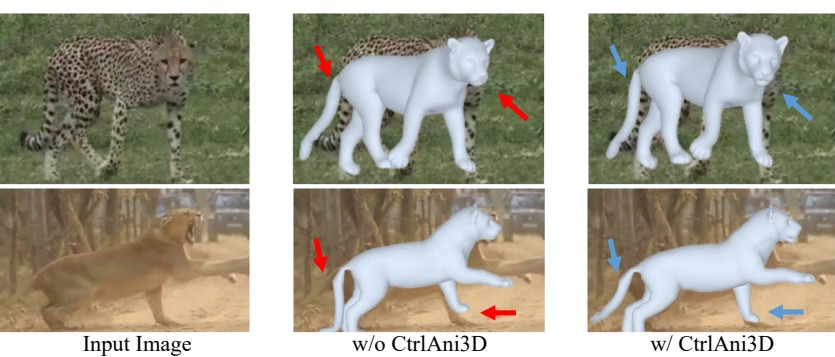

| Input Image | w/o CtrlAni3D | w/ CtrlAni3D |

Figure 6: **Effect of CtrlAni3D on Animal Kingdom datasets.** Input and result images are zoom-in cropped for visualization. Training with CtrlAni3D enhances the model's ability to align tails, limbs and faces. Red arrows indicate misalignments, while blue arrows indicate better alignments.

## 6 CONCLUSION

This paper presents AniMer, a simple yet effective method for accurate animal pose and shape estimation. The key to the success of AniMer is a large capacity Transformer backbone together with an aggregated large scale dataset. Within the aggregated dataset, we propose a novel synthetic general quadruped dataset CtrlAni3D, which is rendered by prompting a controllable text-to-image generation model ControlNet. Benefiting from our design philosophy, AniMer not only outperforms previous methods for general quadruped mesh recovery, but also beats state-of-the-art single-species reconstruction methods such as BITE for dog. We believe the principles behind AniMer would inspire the mesh recovery tasks of the whole animal kingdom, and enable several down-stream applications such as avatar creation and behavioral analysis.

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

# A APPENDIX

## A.1 ABOUT CTRLANI3D

During the generation of CtrlAni3D dataset, we prompt the ControlNet using common names instead of scientific names of animals. However, to better indicate the position our CtrlAni3D dataset in the animal taxonomy, we list the most relevant scientific names of used animal species in Tab. 5.

Table 5: **Scientific names of used animal species in CtrlAni3D.** The size of generated images of each species is listed at the right column.

| Family | Species | Prompt Commands | Count |
|---|---|---|---|
| *Felidae* | *Felis catus* | Cat | 80 |
| | *Panthera leo* | Lion | 630 |
| | *Acinonyx jubatus* | Cheetah | 299 |
| | *Panthera tigris* | Tiger | 280 |
| *Canidae* | *Canis lupus familiaris* | Dog | 2976 |
| | *Canis lupus* | Wolf | 413 |
| *Equidae* | *Equus ferus caballus* | Horse | 2228 |
| | *Equus zebra* | Zebra | 1460 |
| *Bovidae* | *Bos taurus* | Cow | 890 |
| *Hippopotamidae* | *Hippopotamus amphibius* | Hippo | 455 |
| Total | | | 9711 |

## A.2 DESCRIPTIONS FOR FULL TRAINING DATASET

**Animal Pose dataset.** The Animal Pose dataset (Cao et al., 2019) includes five categories: dog, cat, cow, horse and sheep, comprising a total of over 6,000 instances across more than 4,000 images. Each animal instance in Animal Pose dataset is annotated with 20 keypoints.

**APT-36k dataset.** The APT-36k dataset (Yang et al., 2022b) contains 36000 images covering 30 different animal species from different scenes. There are typically 17 keypoints labeled for each animal instance.

**AwA Pose dataset.** The AwA Pose dataset (Banik et al., 2021) is introduced for 2D quadruped animal pose estimation. AwA contains 10064 images of 35 quadruped animal species and each image is annotated with 39 keypoints.

**Stanford Extra dataset.** The Stanford Extra dataset (Biggs et al., 2020) consists of 20,580 images and covers 120 dog breeds. Each image is annotated with 20 2D keypoints and silhouette.

**Zebra synthetic dataset.** The Zebra synthetic dataset (Zuffi et al., 2019) consists of 12850 images. Each image is randomly generated that differs in background, shape, pose, camera, and appearance.

**Animal Kingdom dataset.** The Animal Kingdom dataset (Ng et al., 2022) includes a diverse range of animal species across 8 major animal classes. We only use the part of pose estimation dataset to evaluate our method.

**Animal3D dataset.** Animal3D dataset (Xu et al., 2023a) contains a total of 3379 images, which are classified into 40 classes. Each image is annotated with SMAL (Zuffi et al., 2017) parameters, 2d keypoints, 3d keypoints and mask.

**CtrlAni3D dataset.** Our dataset is annotated the same as Animal3D dataset. Fig. 1 shows overview of our dataset. More details about our dataset can be found in Sec. 4. We randomly set train/test part by a ratio 0.85/0.15.

By aggregating all above datasets for training, we set different sampling weights to different datasets according to the dataset type and dataset size, see Tab. 6.

Table 6: **Full dataset statistics for training.**

| Dataset | Number | Ratio | Training Sample Weight |
|---|---|---|---|
| Animal3D | 3065 | 7.4% | 1 |
| CtrlAni3D | 8277 | 20.0% | 0.5 |
| Animal Pose | 1680 | 4.0% | 0.15 |
| AwA-Pose | 2884 | 7.0% | 0.15 |
| Zebra Synthetic | 12850 | 31.1% | 0.05 |
| Stanford Extra | 7689 | 18.6% | 0.15 |
| APT-36K | 4887 | 11.8% | 0.15 |
| Total | 41332 | 100% | - |

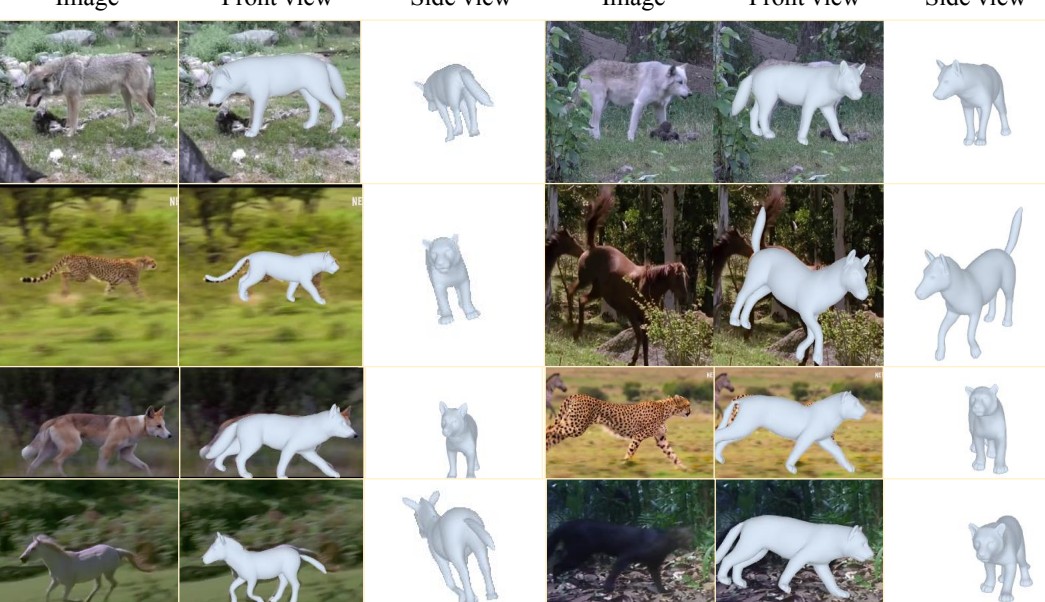

Figure 7: **Results on the Animal Kingdom dataset.** For each case, we display the input image and the output result including a front view rendering and a side view rendering.

### A.3 ADDITIONAL RESULTS

We provide quantitative results on the Animal Kingdom dataset in Fig. 7. Even on these challenging, in-the-wild images, AniMer still achieves reconstructions that align well with the images. This demonstrates the robustness of AniMer. We also provide more qualitative comparisons in Fig. 8. To better compare with Animal3D, we compare qualitative results with it on the result samples presented in their paper (Xu et al., 2023a). Because they have not released their synthetic datasets, we directly take their images and attach our results aside, see Fig. 9. Moreover, to highlight the comparison of CtrlAni3D and the synthetic dataset proposed in Xu et al. (2023a), we follow their strategy to first pretrain HMR on CtrlAni3D for 100 epochs and then on Animal3D for 1000 epochs, see Tab.A.3. For more comparison, 200-epoch pretraining result is also attached.

In Table. 7, Table. 9, Table. 10, Table. 8 and Table. 11, we provide additional ablation study experiments and their respective metric indicators. When training with Animal3D and CtrlAni3D separately, they performed well on their respective evaluation sets. However, their performance did not surpass the results obtained from the full training set. We believe that each training dataset can effectively enhance the model's performance.

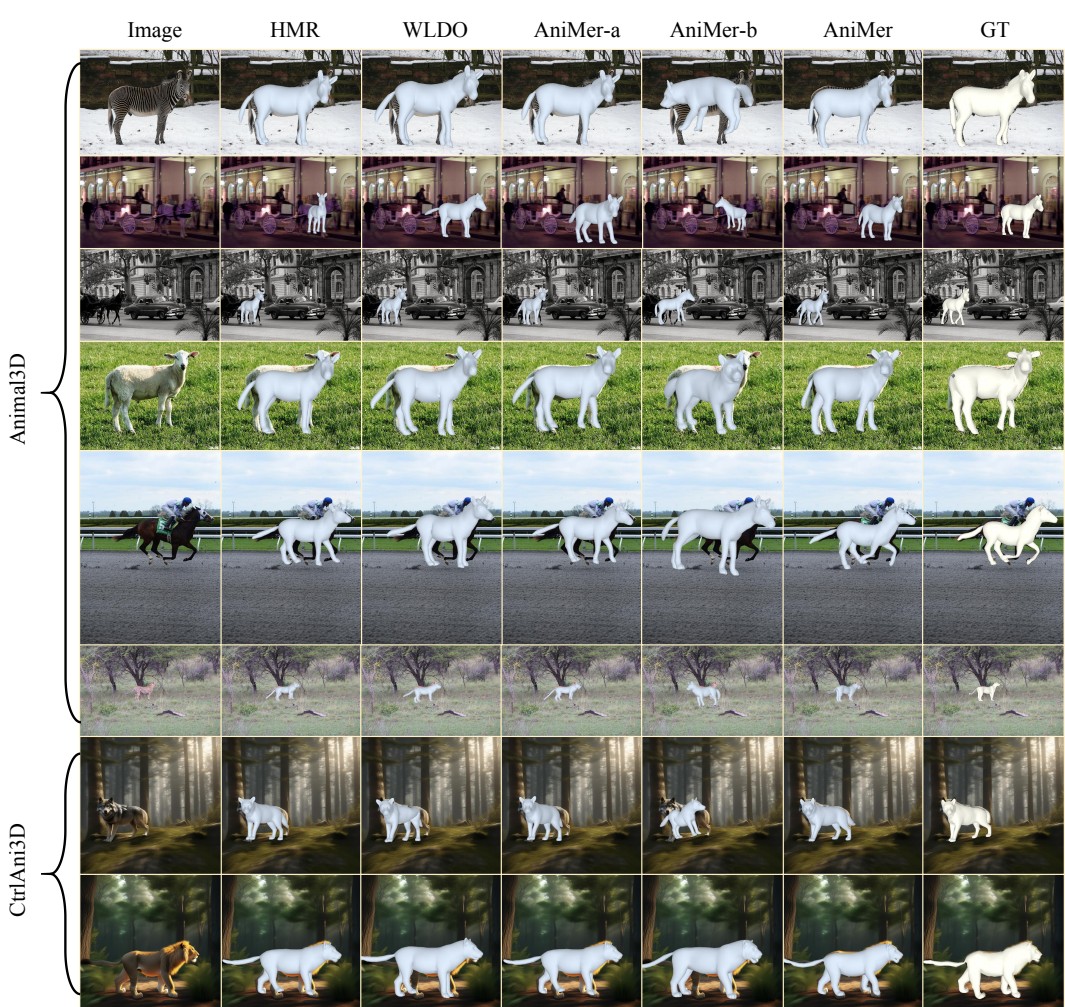

Figure 8: **More qualitative results.**

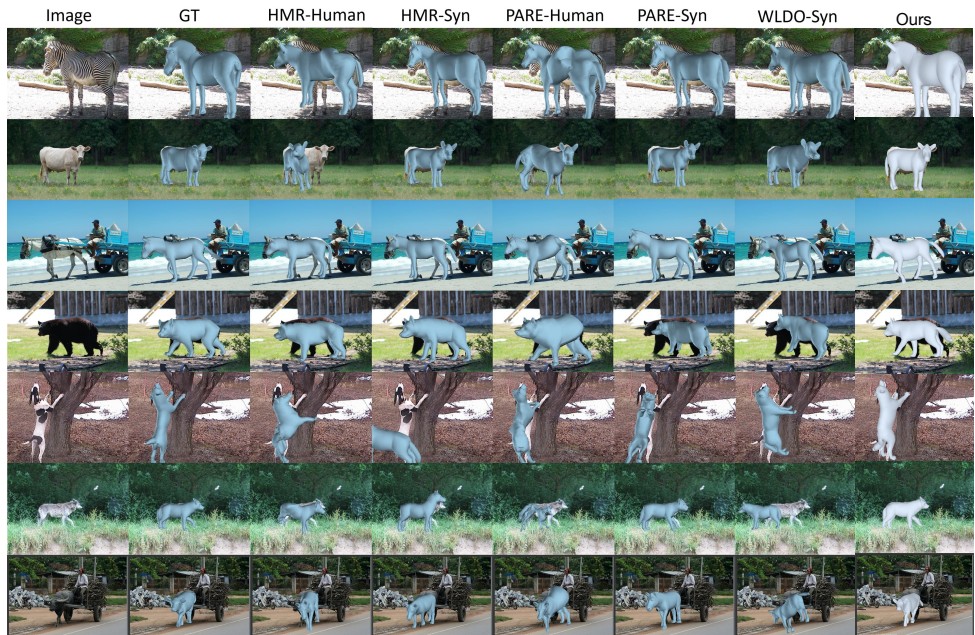

Figure 9: Qualitative results of Animal3D dataset. The last column is our qualitative results and the others is from Animal3D (Xu et al., 2023a) paper.

Table 7: **Full Ablation study result on Animal3D dataset.** Performance of models on Animal3D dataset when trained on the original Animal3D dataset (A3D), our CtrlAni3D dataset (C3D), 2D supplementary datasets, or their combinations. We report PA-MPJPE (PA-J) and PA-MPVPE (PA-V) in millimeter as 3D metrics, and AUC, PCK@HTH, PCK@0.1, PCK@0.15 in percentage as 2D metrics. **Bold** numbers indicate the best performance.

| Animal3D | Training Data | | | 3D Metric | | 2D Metric | | | |
|---|---|---|---|---|---|---|---|---|---|
| ExpNo. | A3D | C3D | 2D | PA-J↓ | PA-V↓ | AUC↑ | P@HTH↑ | P@0.1↑ | P@0.15↑ |
| 1 | ✓ | | | 87.3 | 93.2 | 86.3 | 83.6 | 43.4 | 67.1 |
| 2 | | ✓ | | 109.7 | 115.4 | 82.6 | 78.7 | 32.3 | 54.5 |
| 3 | ✓ | ✓ | | 82.5 | 88.0 | 88.0 | 88.2 | 52.6 | 75.9 |
| 4 | ✓ | | ✓ | 82.0 | 86.5 | 88.7 | 89.2 | 56.8 | 78.6 |
| 5 | | ✓ | ✓ | 110.7 | 117.1 | 87.2 | 85.5 | 49.2 | 71.4 |
| 6 | ✓ | ✓ | ✓ | **81.2** | **85.3** | **89.1** | **89.9** | **59.4** | **79.8** |

Table 8: **Full Ablation study result on AnimalPose dataset (dogs only).** Performance of models on AnimalPose dataset when trained on the original Animal3D dataset (A3D), our CtrlAni3D dataset (C3D), 2D supplementary datasets, or their combinations. We report AUC, PCK@HTH, PCK@0.1, PCK@0.15 in percentage as 2D metrics. **Bold** numbers indicate the best performance.

| AnimalPose* | Training Data | | | 2D Metric | | | |
|---|---|---|---|---|---|---|---|
| ExpNo. | A3D | C3D | 2D | AUC↑ | PCK@HTH↑ | PCK@0.1↑ | PCK@0.15↑ |
| 1 | ✓ | | | 82.1 | 85.7 | 25.4 | 48.2 |
| 2 | | ✓ | | 79.6 | 80.6 | 27.7 | 48.3 |
| 3 | ✓ | ✓ | | 83.9 | 88.8 | 36.2 | 57.7 |
| 4 | ✓ | | ✓ | 86.3 | 91.7 | 43.9 | 67.0 |
| 5 | | ✓ | ✓ | 85.6 | 89.2 | 46.4 | 65.2 |
| 6 | ✓ | ✓ | ✓ | **87.2** | **92.8** | **48.0** | **69.4** |

Table 9: **Full Ablation study result on our CtrlAni3D dataset.** Performance of models on CtrlAni3D dataset when trained on the original Animal3D dataset (A3D), our CtrlAni3D dataset (C3D), 2D supplementary datasets, or their combinations. We report PA-MPJPE (PA-J) and PA-MPVPE (PA-V) in millimeter as 3D metrics, and AUC, PCK@HTH, PCK@0.1, PCK@0.15 in percentage as 2D metrics. **Bold** numbers indicate the best performance.

| CtrlAni3D | Training Data | | | 3D Metric | | 2D Metric | | | |
|---|---|---|---|---|---|---|---|---|---|
| ExpNo. | A3D | C3D | 2D | PA-J↓ | PA-V↓ | AUC↑ | P@HTH↑ | P@0.1↑ | P@0.15↑ |
| 1 | ✓ | | | 90.1 | 95.8 | 88.8 | 88.8 | 53.6 | 79.5 |
| 2 | | ✓ | | 49.7 | 53.9 | 93.5 | **95.4** | 83.6 | 94.2 |
| 3 | ✓ | ✓ | | 54.6 | 59.2 | 92.8 | 95.1 | 80.2 | 93.2 |
| 4 | ✓ | | ✓ | 88.0 | 92.2 | 91.5 | 92.6 | 74.0 | 89.7 |
| 5 | | ✓ | ✓ | 41.2 | 44.7 | 93.9 | 95.4 | 85.4 | **94.7** |
| 6 | ✓ | ✓ | ✓ | **39.4** | **42.7** | 93.9 | 95.3 | **85.6** | 94.3 |

Table 10: **Full Ablation study result on AnimalKingdom dataset. (8 classes)** Performance of models on AnimalKingdom dataset when trained on the original Animal3D dataset (A3D), our CtrlAni3D dataset (C3D), 2D supplementary datasets, or their combinations. We report AUC, PCK@HTH, PCK@0.1, PCK@0.15 in percentage as 2D metrics. **Bold** numbers indicate the best performance.

| AnimalKingdom | Training Data | | | 2D Metric | | | |
|---|---|---|---|---|---|---|---|
| ExpNo. | A3D | C3D | 2D | AUC↑ | PCK@HTH↑ | PCK@0.1↑ | PCK@0.15↑ |
| 1 | ✓ | | | 80.1 | 78.4 | 25.9 | 46.2 |
| 2 | | ✓ | | 76.7 | 77.7 | 24.1 | 40.3 |
| 3 | ✓ | ✓ | | 81.4 | 81.1 | 30.4 | 48.9 |
| 4 | ✓ | | ✓ | 82.3 | 83.6 | 34.2 | 54.5 |
| 5 | | ✓ | ✓ | 83.3 | 84.5 | 35.5 | 55.1 |
| 6 | ✓ | ✓ | ✓ | **83.6** | **85.2** | **37.4** | **57.2** |

Table 11: **Full Ablation study result on STANFORDEXTRA dataset (dogs only).** Performance of models on STANFORDEXTRA dataset when trained on the original Animal3D dataset (A3D), our CtrlAni3D dataset (C3D), 2D supplementary datasets, or their combinations. We report AUC, PCK@HTH, PCK@0.1, PCK@0.15 in percentage as 2D metrics. **Bold** numbers indicate the best performance.

| STANFORDEXTRA* | Training Data | | | 2D Metric | | | |
|---|---|---|---|---|---|---|---|
| ExpNo. | A3D | C3D | 2D | AUC↑ | PCK@HTH↑ | PCK@0.1↑ | PCK@0.15↑ |
| 1 | ✓ | | | 84.7 | 90.1 | 39.6 | 64.1 |
| 2 | | ✓ | | 84.6 | 89.2 | 40.9 | 65.0 |
| 3 | ✓ | ✓ | | 86.6 | 92.4 | 49.8 | 73.0 |
| 4 | ✓ | | ✓ | 88.5 | 94.1 | 61.9 | 81.1 |
| 5 | | ✓ | ✓ | 88.7 | 94.1 | 62.7 | 81.6 |
| 6 | ✓ | ✓ | ✓ | **89.3** | **94.8** | **66.3** | **84.2** |

Table 12: **Effective of CtrlAni3D.** "HMR-Pretrained-Synthetic*" indicates that results are borrowed from Animal3D (Xu et al., 2023a). "HMR-Pretrained-CtrlAni3D-Epoch100" represents that we pretrain HMR for 100 epochs before training it on the Animal3D dataset. Similarly, "HMR-Pretrained-CtrlAni3D-Epoch200" represents that we pretrain HMR for 200 epochs.

| Method | Animal3D | |
|---|---|---|
| | PCK@HTH↑ | PA-MPJPE↓ |
| HMR | 60.5 | 127.8 |
| HMR-Pretrained-Synthetic* | 63.1 | 124.8 |
| HMR-Pretrained-CtrlAni3D-Epoch100 | 64.0 | 121.9 |
| HMR-Pretrained-CtrlAni3D-Epoch200 | **66.1** | **119.6** |

