# OpenReview forum: "AniMer: Animal Pose and Shape Estimation Using Transformer"
_ICLR.cc/2025/Conference — ICLR 2025 Conference Withdrawn Submission_

### Official Review · Reviewer_3uEQ · 2024-10-30

**Soundness:** 3
**Presentation:** 3
**Contribution:** 3
**Rating:** 6
**Confidence:** 4

**Summary:**

The paper introduces a Transformer-based approach for estimating the pose and shape of various quadrupedal animal species, called AniMer. This system aims to address the limitations of traditional methods that focus on specific species and have low network capacity. AniMer leverages a high-capacity Transformer backbone to improve accuracy across diverse species and uses a newly proposed dataset, CtrlAni3D, for training. CtrlAni3D is a large-scale, synthetic dataset created using a diffusion-based image generation model, containing over 9.7k pixel-aligned SMAL mesh-labeled images. By combining AniMer's robust architecture with this expansive dataset, the paper demonstrates performance improvements in animal pose and shape estimation, on both multi-species and single-species benchmarks.

**Strengths:**

+ AniMer uses a high capacity Transformer-based method for estimating poses in various four-legged animals. The Transformer backbone is trained through creating a large synthetic dataset called CtrlAni3D.
+ The authors didn't just test their model on one dataset; they used several—like Animal3D, CtrlAni3D, and Animal Kingdom. This thorough testing shows that their model is robust and works well across different scenarios. The detailed experiments and analyses provide strong evidence of the model's effectiveness.
+ The paper is written clearly and it is easy to follow, including how the Transformer-based approach works and how they generated their dataset. The figures are also informative.
+ This line of work could have a big impact on studies of animal movement and welfare.

**Weaknesses:**

- While their synthetic CtrlAni3D dataset is extensive, relying mainly on synthetic data raises questions about how well the model applies to real-world situations. Testing it on more diverse, real-world datasets could strengthen their claims.
- The model focuses mostly on four-legged animals. It would be interesting to see if it can be applied to other types of animals, like birds or primates.
- Using high-capacity Transformers and large datasets means the model requires significant computational resources. This could make it harder for smaller research teams to use. Exploring lighter versions of the model or providing benchmarks for smaller setups could make it more accessible.

**Questions:**

1. Can the model be adjusted to work with animals that aren't four-legged, like primates or birds? If yes, what challenges might arise in doing so?
2. How does the model perform when tested on real-world images of animals that aren't part of the current benchmark datasets?
3.Are there plans to develop lighter versions of AniMer for use in situations where computational resources are limited?

---

### Official Review · Reviewer_m49W · 2024-11-02

**Soundness:** 2
**Presentation:** 3
**Contribution:** 1
**Rating:** 5
**Confidence:** 5

**Summary:**

The paper proposes AniMer, a Transformer-based model designed to address the challenge of accurately estimating the pose and shape of various quadrupedal animals in a unified structure. Unlike CNN-based methods, AniMer leverages high capacity backbone to generalize well across species and complex poses, directly decoding parameters for the SMAL model to capture detailed animal shapes. To support this, the authors introduce CtrlAni3D, a large-scale synthetic dataset generated with diffusion methods, which provides diverse training data with pixel-aligned SMAL mesh labels. Experiments on Animal3D, CtrlAni3D, and the unseen Animal Kingdom dataset demonstrate that AniMer outperforms CNN baselines in pose and shape estimation, showing strong generalization and robustness,

**Strengths:**

1.AniMer model proposed in this paper enhances the accuracy of general quadrupedal animal pose and shape estimation models. This is a challenging and under-explored field that requires further research, and this work can aid in advancing studies of animal behavior.
2.The introduction of the CtrlAni3D dataset, generated using a diffusion-based pipeline, is a unique contribution that provides synthetic data with pixel-aligned SMAL mesh labels, enriching the training data available for animal modeling and enhancing generalization across species.
3.The paper demonstrates rigorous experimentation with evaluations on Animal3D, CtrlAni3D, and the Animal Kingdom datasets. Performance testing across multi-species and single-species settings shows a thorough evaluation of AniMer's generalization capability. Results indicate that AniMer performs well across all datasets, providing evidence that the model along with CtrlAni3D, improves pose and shape estimation performance over previous methods.

**Weaknesses:**

1. Model Design: The model lacks originality and network architecture looks quite similar to HMR2.0, from input to ViT encoder to feature to Transformer to MLP to SMPL/SMAL paramter. It appears that this paper simply combines HMR2.0 with Animal3D, replacing SMPL with SMAL.
2. Use of Transformer-Based Methods: Although this paper claims to be the first transformer-based method for animal pose and shape estimation, similar questions have likely been addressed in transformer-based methods for general animal pose estimation 2–3 years ago. Strictly speaking, the efficacy of high-capacity networks across multiple animal species has already been demonstrated in prior work.
3.Transformer Decoder Usage: The paper states it used a transformer decoder but does not validate this choice in the ablation study. The only support provided is the statement, "choose to directly decode the final SMAL parameters due to limited SMAL pose prior," which is insufficient. Generally, modifications of previous work require a detailed explanation. The authors should conduct ablation experiment to validate the mechanism proposed in the paper.
4.Two-Stage Training: Another change compared to HMR2.0 is the two-stage training method. However, there is no supporting content or ablation study to validate this choice. For original contributions, novel modifications need to be experimentally verified, or citations should be provided if the changes stem from others' work. The same to 3, an ablation experiment with analyses that would effectively validate the two-stage training approach.
5.CtrlAni3D Dataset Creation: One of the contributions is the creation of the CtrlAni3D dataset, but there are several aspects of its generation process that remain unclear. For example, how were backgrounds provided and what prompts were used? According to SPAC-Net: Synthetic Pose-aware Animal ControlNet for Enhanced Pose Estimation, merely providing general prompts can make it challenging for ControlNet to generate detailed and diverse backgrounds. Additionally, how was the reasonableness of animal poses ensured? Some theoretically possible poses are uncommon in reality, and simply setting a range for uniform sampling lacks justification. The authors can provide more details in the paper to describe the background and reasonable pose generation.
6.Choice of Baselines: For multi-species experiments, WLDO and HMR are both more than five-year-old models, making the experiments less reliable. The authors should add some more recent baseline models that would provide a more meaningful comparison.

**Questions:**

1.The structure of the paper is overly simplistic. Given the widespread use of ViT in pose estimation, numerous derivative works have been developed, such as transformers with depth, transformer-graph hybrid architectures, and dynamic fusion modules. The authors could explore further structural optimization to improve the model's performance.
2.According to Weaknesses 3 and 4, authors should add more verfication and ablation study to support the benefits of the changes.
3.Loss functions such as 𝐿2𝑑 and 𝐿adv, along with others, should be formatted as equations with index in the paper.
4.The motivation section in Section 4 should ideally be fully explained in the introduction.
5.According to Weaknesses 5, It's better for authors to give more explanation on data generation, especially on how to generate reasonable poses.
6.The reason for omitting common metrics like MPJPE is unclear.
7.Single-species experiments lack visual results from other methods for comparison.
8.In the ablation study for CtrlAni3D, why is A3D always included in the training data? This setup may not effectively demonstrate the superiority of the C3D dataset over previous A3D datasets.

---

### Official Review · Reviewer_ze3Z · 2024-11-02

**Soundness:** 2
**Presentation:** 3
**Contribution:** 2
**Rating:** 3
**Confidence:** 4

**Summary:**

In this work the authors proposed AniMer, a transformer model that estimates animal pose and shape. AniMer modifies the residual parameter decoding and adopts a two-stage training scheme. Moreover the authors proposed a CtrlAni3D dataset that generates diverse annotated animal images with pretrained diffusion models.

**Strengths:**

1. With a combination of model design and dataset construction, AniMer demonstrated improved quantitative performance and qualitative results. With a simple transformer architecture design and a scaling dataset, this recipe will enable more valuable research and models for animal pose estimation.
2. The proposed CtrlAni3D dataset is well constructed with chatgpt-generated prompts and semi-automated filtering. The quality of the dataset allows AniMer to scale better.

**Weaknesses:**

1. Are there ablation study results on the two model designs (L208-215)? I don't find it in the main paper or appendix.
2. The experiments are conducted on in-distribution testing data only. It remains unclear how the model would perform in cases with partial occlusion, truncation, or out-of-distribution poses. This is particularly important as such OOD-cases are often very common in the wild.
3. Does the cycle consistency lead to a trade-off between diversity and quality? E.g., certain poses, prompts are harder to generate, and more likely to be filtered?
4. [A] showed that images generated by 3D-conditioned diffusion models demonstrate artifacts and domain gap with real images, especially in out-of-distribution poses. Does the authors observe similar issues? If so, how can these cases be avoided?
5. I acknowledge the importance of the work and the value of AniMer. However this work lacks technical novelty. Specifically synthetic images with 3D annotations have been widely adopted in previous works.

[A] Generating Images with 3D Annotations Using Diffusion Models

**Questions:**

1. What are other potential limitations of the approach or the AniMer model other than **[W2]**?
2. Can the authors present some qualitative examples of the CtrlAni3D dataset, specifically success and failure cases?

---

### Official Review · Reviewer_npgS · 2024-11-03

**Soundness:** 3
**Presentation:** 3
**Contribution:** 3
**Rating:** 6
**Confidence:** 3

**Summary:**

The main contribution is (1) the introduction of a transformer-based model for animal pose and shape estimation, (2) scaling training set by aggregating existing 2D and 3D datasets as well as curating a large synthetic dataset using conditional diffusion model. The transformer model follows HMR 2.0 closely, which consists of a ViT backbone, a decoder and a lightwight MLP that regresses the parameters of SMAL model. On 3 animal datasets, the proposed architecture shows significant improvement over baselines, showing effectiveness of the transformer-based architecture and the dataset scaling.

**Strengths:**

- Good performance. This is a system paper that focuses on the training recipe to maximize performance. Although the architecture is not novel and follows the architecture of HMR 2.0 closely, the significant performance improvement on the animal dataset itself is a contribution.
- Well-motivated design choices. The paper carefully explains the motivation behind certain design choices and training strategies to account for the lack of animal pose prior as compared to the human pose prior, and the limited quantity of animal 3D datasets. These insights are helpful.

**Weaknesses:**

- Insufficient baselines and ablations.
  - This paper is a animal version of HMR 2.0, with a small difference in the MLP part: the authors replaced the iterative refinement with directly regressing, because the lack of SMAL mean priors. However, in the experiments the authors only included 2 ablated models with different backbones. I think at least the authors should include the following to justify the design choices made in the paper: (1) the proposed model with HMR 2.0's MLP. (2) the proposed model with HMR 2.0's backbone. (2) without the two-stage training strategy.
  - In Table 3, the paper looks at the effect of each training set, and shows the effect of using A3D, A3D+C3D, and A3D+C3D+2D. However, I think the authors should include A3D+2D numbers, because A3D+2D are existing 3D and 2D datasets, and C3D is the proposed dataset. It makes sense to report the improvement brought by this new dataset.

**Questions:**

- The ViT encoder is pretrained on ImageNet using MAE, whereas in HMR 2.0 the ViT backbone is pretrained on human datasets. I understand that because there is a limited amount of animal datasets, it does not make sense to pretrain on them. But I wonder if the authors have tried using ViTPose as backbone, to see if pretraining on human datasets with key point loss would help with animal shape and pose prediction
- Line 95: The curation of CtrlAni3D requries manual verification. Ln 299: "we manually filter out images that do not match the mesh poses to ensure good data quality.” A few examples would be helpful.

---

### Note · Authors · 2024-11-13

**Comment:**

Thanks very much for the efforts of all reviewers! They all provide constructive feedbacks, especially on the ablation studies, details about dataset construction, and technical novelty. Finally, we are withdrawaling to make some substantial improvements over the technical pipelines and evaluations, which may not be fully addressed by only discussion. We thank again for all the reviewers for their insightful comments, we will carefully answer all the questions in our next version.

**Withdrawal Confirmation:**

I have read and agree with the venue's withdrawal policy on behalf of myself and my co-authors.